# Osteosarcoma Multi-Omics Landscape and Subtypes

**DOI:** 10.3390/cancers15204970

**Published:** 2023-10-13

**Authors:** Shan Tang, Ryan D. Roberts, Lijun Cheng, Lang Li

**Affiliations:** 1College of Pharmacy, The Ohio State University, Columbus, OH 43210, USA; tang.1318@osu.edu; 2Department of Biomedical Informatics, College of Medicine, The Ohio State University, Columbus, OH 43210, USA; lijun.cheng@osumc.edu; 3Department of Pediatrics, College of Medicine, The Ohio State University, Columbus, OH 43210, USA; ryan.roberts@nationwidechildrens.org

**Keywords:** osteosarcoma, bioinformatics, multi-omics, clinical outcome, somatic copy-number alteration, gene expression, methylation

## Abstract

**Simple Summary:**

Molecular subtyping and therapeutic target identification in osteosarcoma have been extremely challenging due to the high degree of heterogeneity in its genetic profile and the lack of specific oncogenic driver genes. Integrating multiplatform profiles of somatic copy-number alteration, gene expression and methylation, three molecularly distinct and clinically relevant subtypes of osteosarcoma were revealed. Additionally, several novel gene signatures were identified based on the subgrouping results. We demonstrated the pivotal role of the cross-platform analysis for investigating complex diseases such as osteosarcoma.

**Abstract:**

Osteosarcoma (OS) is the most common primary bone malignancy that exhibits remarkable histologic diversity and genetic heterogeneity. The complex nature of osteosarcoma has confounded precise molecular categorization, prognosis, and prediction for this disease. In this study, we performed a comprehensive multiplatform analysis on 86 osteosarcoma tumors, including somatic copy-number alteration, gene expression and methylation, and identified three molecularly distinct and clinically relevant subtypes of osteosarcoma. The subgrouping criteria was validated on another cohort of osteosarcoma tumors. Previously unappreciated osteosarcoma-type-specific changes in specific genes’ copy number, expression and methylation were revealed based on the subgrouping. The subgrouping and novel gene signatures provide insights into refining osteosarcoma therapy and relationships to other types of cancer.

## 1. Background

Osteosarcoma (OS) is a primary bone malignancy, and the most common bone cancer in pediatric patients. The standard of care for all patients diagnosed with osteosarcoma remains MAP chemotherapy, the same treatment first introduced in the late 1980s [1]. Osteosarcoma has historically been sub-grouped into osteoblastic, chondroblastic, fibroblastic, small cell, telangiectatic, high-grade surface, extra-skeletal, and other lower-grade forms [2]. Whether the histologic appearance of these subgroups reflects differences in the cell of origin or of distinct molecular drivers of oncogenesis remains unclear. Although several groups have attempted to use histology to guide therapy, no systematic approach to subgrouping has proved clinically useful. The pervasive and extensive structural rearrangements characteristic of osteosarcoma tumors further complicates the identification of molecular features that are associated with osteosarcoma pathogenesis.

Molecular genetic studies of osteosarcoma have primarily focused on specific genes or pathways using single platform data. Single nucleotide polymorphism (SNP), structural variations (SV), somatic copy-number alterations (SCNAs), loss of heterozygosity (LOH), and RNA expression have all been investigated [3,4]. These molecular studies have yielded insights, finding driver genes such as p53, Rb1, CDKN2A and PTEN, and aberrant signaling in the PI3K/mTOR, IGF, and other signaling pathways [1,2,3,5,6]. Noticeably, a higher presence of SCNA, a form of genomic structural variation that often affects large segments of the genome (even whole chromosomes), is reported in osteosarcoma compared to SNPs and other mutations [7], suggesting a crucial role for SCNA in gene regulation and pathobiological process in osteosarcoma. Kovac and others have also suggested that these complex SCNA patterns in osteosarcoma develop early in the oncogenic process but persist stably, both before and after the detection of primary tumors [8,9]. To address the inter-patient and intra-tumoral heterogeneity of osteosarcoma, integration analysis of multi-omics data, including genomics, transcriptomics, proteomics and metabolomics, has emerged to provide a more comprehensive view of biology. For example, Futreal conducted whole genome, RNA, and T-cell receptor sequencing, immunohistochemistry and reverse phase protein array profiling (RPPA) on 48 osteosarcoma specimens to characterize the immuno-genomic landscape in osteosarcoma [10]. The international TARGET (Therapeutically Applicable Research to Generate Effective Therapy) osteosarcoma project has been a valuable source for integrative genomic analyses. TARGET osteosarcoma molecular characterization analyses include gene expression array, copy number array, methylation, whole genome sequencing, whole exome sequencing, miRNA-seq, and mRNA-seq. 

Previously, the potential prognostic biomarkers and subtyping in osteosarcoma was mainly identified from single-platform data and a clustering approach. For example, Southekal et al. [11], recently found several prognostic biomarkers using expression data of miRNA, mRNA, and lncRNA from the TARGET dataset. They also revealed two subtypes of osteosarcoma, using clustering and Cluster-Of-Clusters Analysis (COCA) for gene expression, DNA methylation, and miRNA expression. Different from the traditional clustering approach, we applied similarity network fusion (SNF) to multi-omics data in osteosarcoma samples. The SNF substantially outperforms single data type analysis and clustering analysis based on this study and previous research [12]. The performance comparison between integrated multi-omics data and individual omics data has not been covered by Southekal et al. In addition to innovation in methodology, the combination of SCNA, gene expression and methylation was used as a SNF input in our study. The progressive recognition of SCNA as a driving factor in osteosarcoma pathogenesis has been discussed [7]. The importance of methylation in cancer research, especially for subtyping and biomarker selection, has also been summarized [13]. A network assembling well-known and lesser-known candidates for osteosarcoma was constructed by Poos et al. from only SCNA and expression data (no methylation) [14]. Most importantly, our subgrouping outcomes have been shown to be closely related to clinical outcomes and were further validated in another cohort of osteosarcoma tumor samples. In Siddesh and Pekarek’s studies as mentioned above [11,15], there was no validation of the network module or subtyping criteria, and no discussion about clinical outcome association in either of the studies. Pekarek et al. recently evaluated the serological, genetic and other biomarkers in osteosarcoma, of which the implementation is limited in clinical practice [15]. 

In summary, we used multi-platform profiling and SNF on osteosarcoma patient samples with diverse clinical outcomes. SCNAs, gene expression and methylation data were combined, seeking to bring novel insights into the biological process underlying osteosarcoma tumors that have distinctly different prognosis. 

## 2. Method

### 2.1. Data Collection

The data information is listed in Appendix A. The raw files from the Therapeutically Applicable Research to Generate Effective Treatments (TARGET) data, including the platforms of Affymetrix Exon ST, Affymetrix SNP 6.0, and mRNA-seq, were downloaded via Globus. For the validation set, we downloaded the raw files from ArrayExpress under the accession of E-MTAB-3034 and A-AFFY-142. The corresponding clinical data is kindly provided by Dr. Jan Smida and Dr. Eberhard Korsching. The patient follow-up time is up to 16 years based on the available patients’ clinical data from TARGET and the validation data set. The methylation profiles of two normal human bone samples were obtained at Gene Expression Omnibus (GEO) with the accession of GSE125645.

We downloaded relevant drug sensitivity data from the Genomics of Drug Sensitivity in Cancer (GDSC, version 17.3). This data set includes drug response data and genomic marker genes of sensitivity.

### 2.2. SCNA Detection

We used Birdsuite 1.5.5 (Broad Institute, Merkin Building, 415 Main St., Cambridge, MA 02142) [16] and PennCNV [17] to detect SCNA from Affymetrix Genome-Wide Human SNP Array 6.0. The Canary from Birdsuite, a one-dimensional GMM clustering method, determines the copy number (0, 1, 2, 3, 4) at each predefined CNP locus. Additionally, Birdseye from Birdsuite and PennCNV implement hidden Markov model (HMM) algorithm to infer rare SCNA calls and provide specific copy number of the chromosome region for individual genotyped samples. Then, the regions with copy number information were annotated by HGNC gene symbol and Entrezgene ID from the Ensembl database. After annotation, we combined the SCNA results from Birdsuite (Canary calls surpassing confidence threshold 0.1, and Birdseye calls surpassing a cutoff of 5) and PennCNV to get SCNA for each sample at the gene level. The status of genes was considered to be a deletion when the copy number was 0 or 1, and as an amplification when the copy number was 3 or 4.

### 2.3. Gene Expression

The processed gene expression data from either the GeneChip™ (AFFYMETRIX, INC., 3380 Central Expressway, Santa Clara, CA 95051 USA) Human Exon 1.0 ST Array or the Human Gene 1.0 ST Array was downloaded directly from TARGET dbGAP (accession phs000218.v22.p8.c1.) and ArrayExpress (A-AFFY-142). These gene expression data have been preprocessed as TPM value (transcript per million). To combine datasets from different resources, we further normalized all data using quantile normalization from R preprocessCore package. Batch effects were removed by R sva package with ComBat.

### 2.4. Methylation

The processed and normalized methylation data was downloaded directly from TARGET dbGAP (accession phs000218.v22.p8.c1). For the final matrix, the top one percent of variable probes were retained based on the variation among the samples for further downstream clustering analysis. Then R FDb.InfiniumMethylation.hg19 package was applied to find the closest genes of specific IlmnID for annotation purposes.

### 2.5. SNF

Similarity network fusion (SNF) [12] firstly constructs the networks of samples from each data type separately, then fuses this information into one similarity network. This process takes advantage of each data type, and fully represents the spectrum of underlying data. We used the R package SNFtool (version 2.3.1) to fuse the matrix of SCNA, gene expression and methylation. We used default parameters: K (number of neighbors) = 20, alpha (hyperparameter) = 0.5, T (number of iterations) = 20, for both subgrouping and prediction.

### 2.6. MARINa

MAster Regulator INference algorithm (MARINa) [18] infers transcription factors (TFs) controlling the transition between the two phenotypes, subgroup1 vs. subgroup2 in our case. It performs a Kolmogorov–Smirnov test for each regulator and generates scores for the comparison between two sample groups for each regulator. The R package of ssmarina was specifically applied to conduct the MARINa inference between two subgroups of osteosarcoma patients. The gene expression tables for two subgroups were inputted, the outcome of MARINa was further bootstrapped to increase the accuracy. The top five up-/down-regulated master regulators were further evaluated.

### 2.7. Survival Analysis

We performed survival analysis using the log-rank test and Cox-regression analysis (*p*-value ≤ 0.05) for three subgroups identified by the SNF method, and the patients with different SCNA status on 17p13.1–17q11.2. The R tool, “survival” was used for survival analysis, and the Kaplan–Meier survival curve plots were generated. In addition, a log-rank or Mantel–Haenszel test was conducted to assess survival differences between patients with high or low expression (median as threshold) of the top ten master regulators. Univariate Cox hazard ratios (HRs) with 95% confidence intervals were calculated.

### 2.8. Pathway Analysis

KEGG (Kyoto Encyclopedia of Genes and Genomes) enrichment analysis of a gene set was performed using R clusterProfiler package with Benjamini–Hochberg (BH) adjustment. Then, a dot plot was used for visualization.

### 2.9. Other: Statistical Analysis

Except when specified, all analyses were performed using R (4.0.3). Except for the specified statistical significance in Section 3.3.1 when conducting genome wide analysis of SCNA, gene expression and methylation, a *p*-value or adjusted *p*-value < 0.05 was considered significant.

The sample’s size has been pre-determined, and after the subgrouping we have three subgroups of the patients with sample sizes of 30, 27, and 34 for the training set. The statistical power was calculated from R package pwr using an alpha level of 0.05, and a moderate effect size of 0.5, which yields power greater than 0.9 for a comparison between any two groups. In the validation set, there were 16, 10, and 8 samples in three groups. Using the same type I error and effect size, the lowest power was 0.5 for subgroup 3 with sample size 8; whereas the other two powers reached 0.85, using the same parameters. This statistical potential has illustrated a high likelihood of detecting meaningful differences by SNF subgrouping.

Unsupervised hierarchical clustering with Euclidean distance was used for the cluster analysis on single set of data, i.e., SCNA, gene expression, and methylation. For relatively moderate sample size or categorical variables, we used non-parametric tests. Specifically, Fisher’s exact test was applied to drug response and metastasis data to evaluate the performance of predictive classification. The Kruskal–Wallis test was applied to find the most varied genes among the three subtypes. When the sample size was large or the variables were continuous, the *t*-test was applied. The Pearson’s correlation coefficient was calculated for the association between gene expression and SCNA. The Pearson correlation measures the strength of the linear relationship between two variables, with a value of −1 meaning a total negative linear correlation, 0 being no correlation, and +1 meaning a total positive correlation.

## 3. Results 

### 3.1. Result 1: Sample and Data Collection

Eighty-six osteosarcoma tumor samples were obtained from the Therapeutically Applicable Research to Generate Effective Treatments (TARGET) program with matched SCNA, gene expression and DNA methylation profiles. Another cohort of 38 osteosarcoma samples from Dr. Eberhard Korsching’s lab [14] with SCNA (E-MTAB-3034) and gene expression (A-AFFY-142, RMA normalized) data were used as a validation set. Clinical and pathologic data are summarized in Table 1. The patients with >90% necrosis at definitive surgery or histologic response stage 3/4 (91–100% necrosis) were annotated as good responders to chemotherapy treatment, while the patients with <90% necrosis at definitive surgery or histologic response stage 1/2 (0–90% necrosis) were annotated as poor responders. Due to the overall rarity of osteosarcoma tumor samples, not all samples have the matched multi-platform data available: in our validation set, the gene methylation data is absent. We included the methylation profiles of two normal bone samples from GSE125645. To ensure the data integrity and consistency, all the selected samples used the same platforms and batch effects were removed. Appendix A shows the data sources, platforms and processing pipelines. 

### 3.2. Result 2: Similarity Network Fusion Analysis Reveals Three Subtypes of Osteosarcoma 

Considering the large total number of genes in the human genome, we firstly filtered the gene data based on the following criteria: genes with abnormal copy number representing at least 30% of samples; genes with either high expression (top 1000) or high variation (top 2000) in expression; genes with high variation in methylation (top 1%). 

Similarity network fusion (SNF) [12] is designed to firstly construct similarity networks of samples from single-platform data sources, and then integrate these networks into a single similarity matrix using a nonlinear combination method. It enables us to capture variability in similarity across diverse platforms by integrating data in the space of samples rather than measurement. We computed and fused the similarity networks from SCNA, expression, and DNA methylation data by SNF, which yielded three separate subgroups with divergent clinical outcomes and genomic features (referred to as SNFgroup1, SNFgroup2 and SNFgroup3). The patients in SNFgroup1 have a significantly better prognosis including better response to chemotherapy, greater chance of survival and lower incidence of metastasis, compared with the other two subgroups. The *p*-values for survival, drug response and metastasis among these three subgroups are summarized in Table 2. We also calculated the Hazard Ratios (HRs) using a Cox proportional-hazards model with SNF group1 as the reference, as shown in the survival plot in Figure 1. To validate our result, the SNF method was applied to label another 38 osteosarcoma patient samples with only matched profiles of SCNA and gene expression in the validation set, based on their similarities with the TARGET cohort in the SCNA and gene expression. Despite the lack of methylation profiles, the predicted subgroups showed similar a pattern of clinical outcomes: patients in the second subgroup have the worst prognosis, while patients in the first subgroup have the best prognosis.

We also performed unsupervised hierarchical clustering, a method widely used in omics data analyses, on single-platform data (Appendix A). We noticed the subgrouping results from unsupervised clustering on single data sources vary from one to another (Appendix A). To further evaluate subgrouping performance on differentiating clinical outcomes, we calculated the *p*-values for survival, drug response and metastasis among three subgroups from SNF or unsupervised clustering (Table 2). The log-rank test was applied to survival data, and Fisher’s exact test was applied to drug response and metastasis data to evaluate the predictive classification model performance. The noticeable *p*-value changes using SNF, suggest combining multiple sources of data substantially outperforms single data type analysis when correlating genomic profiles with clinical characteristics. It can also avoid the inconsistency of subgrouping from different single data sources. 

### 3.3. Result 3: Genomic Landscape of Three Osteosarcoma Subtypes

#### 3.3.1. Individual Genes with Significant Differences in SCNA, Expression and Methylation

After subgrouping, we investigated the most varied genes among these three groups by calculating the Kruskal–Wallis test *p*-values for individual genes across chromosome 1 to 22. For SCNA data, we only focus on genes (4400 in total) that have abnormal copy number represented in at least 30% of the samples to avoid the over-representation of normal copy number, which might influence the statistical significance. The Manhattan plots (Figure 2) show the significance level of each gene associated with subgroups and its location on the chromosome for different data sets. The genes with significant changes in single-profiles of SCNA, expression and methylation among the three subgroups from both training and validation set are summarized in Appendix A, satisfying the conditions: (1) for SCNA, with *p*-value < 0.001 in the training set and *p*-value < 0.05 in the validation set; (2) for expression, with *p*-value < 1 × 10^−5^ in the training set and *p*-value < 0.01 in the validation set; (3) for methylation, with *p*-value < 1 × 10^−6^ for the training set. 

There are 21 genes (listed in the Appendix A) that have *p*-values smaller than 0.01 in SCNA, gene expression and methylation from the training and validation sets, which are highlighted in red in the Manhattan plots (Figure 2). This statistical significance suggested the potential roles of these genes in differentiating the OS patients. Among them, MLLT3 [19], ATM/ATR [13,20,21], DDX10 [22,23,24], CASP1 [6] and BRD4 [25,26] have been extensively studied and shown to be associated with pathogenesis or clinical outcomes in osteosarcoma. Additionally, UBASH3B [27,28], CASP1 [29], CARM1 [30,31,32,33], CHAF1B [34,35] and PCNT [36,37] have been studied in other cancers. For example, overexpression of CARM1, an epigenetic enzyme and co-transcriptional activator, has been reported to predict poor prognosis in breast cancer [33] and non-small cell lung cancer [31]. There are several studies that suggested targeting CARM1 to be an effective therapeutic strategy for several diseases including acute myeloid leukemia [32] and ovarian cancer [30]. These findings have highlighted the potential use of our candidate genes as targets for different subgroups of osteosarcoma patients.

#### 3.3.2. SCNA Profiles

We used BirdSuite [16] and PennCNV [17] to detect SCNA and observed a strong association between SCNA and osteosarcoma cancer outcome. Recurrent SCNAs include losses in 3p12.3, 6p22.1, 17q21.2 and 22q11.22; and gains in 6p12.3, 8p23.1–8q24.3 and 17p11.2-p12 were identified (Figure 3A) as previously reported [3,38]. Osteosarcoma patients with poor response to chemotherapy generally have a significantly higher fraction of abnormal copy number within all chromosomes compared with the good responders (Appendix A), which is consistent with Meltzer’s findings [39]. The patients in the second SNFgroup with the worst clinical outcomes tend to have abnormal gene copy number at known cancer driver genes such as p53, PTEN, PB1 and CDKN2A [2,5]. By investigating SCNA patterns of the marker genes identified in Section 3 and Section 3.1 (*p*-value < 0.001 in the training set and *p*-value < 0.05 in the validation set among the three SNFgroups), we found that a large percentage of patients, ranging from 17% to 52%, are inclined to have abnormal copy number (either gain or loss) for these genes. An overall trend of higher percentage of patients with copy number gain, compared with copy number loss (annotated as “gain%” and “loss%”, respectively, in Appendix A) was observed.

We calculated the Kruskal–Wallis test *p*-values for the 4000 genes used in the SCNA analysis (with abnormal copy number represented in at least 30% of patients) among the three subgroups. The top 50 genes that significantly vary among the three subgroups are mostly located at 17p13.1–17q11.2 (Figure 2A and Figure 3B). The amplification and overexpression of genes in 17p11.2–p12 has been reported in previous studies [40,41]. We further noticed that low total SCNA burden (normal or loss) in this region conferred a statistically significant benefit in overall survival, which is verified in our validation cohort (Figure 3C and Appendix A). Analysis in breast cancer also revealed an association between low total SCNA burden at Chr17 and patients’ overall survival [42]. Interestingly, we observed a strong positive correlation between gene expression and copy number within this region, with COPS3 ranked as the highest. There are several previous studies illustrating the occurrence of COPS3 overexpression and TP53 mutation [43], the link between COPS3 and TP53 protein degradation [44], and the central role of TP53 in osteosarcoma pathogenesis. These findings suggest that gain of COPS3 may result in abnormal TP53 behavior that introduces genomic instability and drives osteosarcoma development. Other genes in the 17p13.1–17q11.2 region include RADS1, CTC1, WRAP53, etc. CTC1 and WRAP53, and have been found to be responsible for maintaining telomeres and genome integrity [45,46]. Higher transcript levels of RADS1 and WRAP53 at diagnosis have been shown to be associated with poor prognosis in ALL [47] and head and neck cancer [48], respectively. Other chromosome regions that are associated with clinical outcomes include 6p24.2, 6p21.1, 1p36.33, and 11q23.1, which are suggested by the large −logPval values in Figure 2A,D. For example, RUNX2 at 6p21.1 has been known to be a driver gene in pathogenesis and chemoresistance development in OS [49,50].

The Pearson’s correlation between SCNA and gene expression for the genes that have at least 30% SCNA across the TARGET samples were calculated. The heatmap of the correlation showed strong relationship between neighboring genes, especially genes within the same chromosome (Figure 4A). The qualitative relationship between genetic variation and its downstream effect, especially for oncogenes and tumor suppressor genes, has been supported in previous studies among various cancers [51,52]. It can be interpreted as evidence for an underlying structural mechanism that links gene expression to the copy number.

Nevertheless, whether SCNA alone is able to translate into changes in gene expression has been questioned. In our analysis, we further ranked the genes according to the Pearson correlation value, with representative oncogenes and tumor suppressor genes highlighted in orange in Figure 4B. The 21 potential maker genes (discussed in Section 3 and Section 3.1) are also included in the figure (highlighted in blue). The correlations between SCNA and gene expression among these oncogenic, suppressive, or potential marker genes are generally positive, but the degrees of the association vary considerably. Previous studies imply that the inconsistency between SCNA and expression can be contributed to by length of chromosomal region with abnormal copy number [53], gene dosage compensation [54], etc. The diverse correlations highlight the heterogeneous nature of osteosarcoma and the need to consider these dynamics for identifying biomarkers with higher specificity.

#### 3.3.3. Gene Regulatory Network Hubs Based on the Transcriptome Data

We used RNA-seq data to profile transcriptional gene expression in these samples from osteosarcoma patients. The algorithms of master regulator inference algorithm (MARINa) [18] is designed to infer transcription factors (TFs) or Master Regulators (MRs) between two phenotypes. To identify genes that have a regulatory effect over other genes and possibly define the diverse prognosis in osteosarcoma, we applied MARINa to the groups of patients with better (SNFgroup1) vs. worse (SNFgroup2) clinical outcomes. The top five MRs, prioritized according to the average dysregulation (normalized enrichment score (NES)) and *p*-value, with differential activity (activation or repression), in SNFgroup1 as compared with SNFgroup2, are shown in Figure 5A. Then, we investigated the survival outcomes of all osteosarcoma patients with higher or lower expression (compared with median expression) of these gene regulators. A hazard ratio above one indicates more hazard and hence worse survival in the higher expression than the lower expression group. Specifically, patients with higher expression in the activated genes such as CDK6 and EGFR tend to have worse survival, whereas lower expression in the regressed genes is associated with better survival. This pattern was validated in another osteosarcoma patient cohort from A-AFFY-142 (Appendix A).

Notably, three transcriptional modules out of the top ten MRs, CDK6 [55], EGFR [56] and PANX3 [57], have already been proven to be critical mediators of pathogenesis in osteosarcoma. Targeting PANX3, for example, can inhibit the tumorigenesis of osteosarcoma [57]. More recent studies indicated CDK6 and EGFR contribute to the development of chemotherapy resistance in osteosarcoma [58,59]. There is an ongoing clinical trial with Abemaciclib, an inhibitor of CDK4/6 for patients with chondrosarcoma and osteosarcoma (NCT04040205). In addition, a majority of the key MRs identified in our study, such as EDIL3, PMEPA1 and SATB2, are associated with epithelial–mesenchymal transition (EMT). EMT and EMT-associated transcription factors have been demonstrated to be closely associated with enhanced drug efflux and slow cell proliferation, contributing to enhance general drug resistance [60]. In osteosarcoma, EMT also plays pivotal roles in tumor formation and malignancy [60]. Lastly, the drug sensitivity data from Genomics of Drug Sensitivity in Cancer (GDSC, version 17.3) showed osteosarcoma cell lines with various response to CDK6 or EGFR inhibitors (Appendix A). A more recent publication indicated that gefitinib, an EGFR inhibitor, reduces osteosarcoma invasion and metastasis in vivo [61]. These findings are compelling evidence for the MRs identified in our study serving as clinical evaluation candidates or additional therapeutic options.

#### 3.3.4. Methylome Landscape of Three Subtypes

The role of DNA methylation, a pivotal form of epigenetic modification in gene regulation, has provided a means of dynamic regulation of gene expression over the relatively static genome. In our study, unsupervised clustering of methylation for subgrouping performs better than the other two single-platform data sets when relating the clustering result to clinical outcomes, as demonstrated by the smaller *p*-values in survival and drug response (Table 2). Figure 6A shows the methylation landscape for individual genes with high variation across all samples (top 1%, the input of methylation data in SNF analysis). The patients in the second subgroup, with worse clinical outcomes, appear to have a higher overall methylation value compared with the other two groups.

Among these top 1% of variable genes in methylation, we further calculated the One-Way ANOVA (ANOVA) *p*-values based on the SNF subgrouping results. KEGG enrichment analysis (BH adjusted, 0.05 *p*-value cutoff and 0.1 q-value cutoff) was then applied to the genes with *p*-values smaller than 0.05 from the ANOVA test (Figure 6B). Hippo and Wnt signaling pathways appear to be the top enriched pathways for these highly variable genes in methylation among the three subgroups. Genes in the Hippo signaling pathways are majorly hypermethylated in the OS tumors. Several key genes in the Hippo signaling pathway, such as TEAD1 and TP73, are significantly hypermethylated in the patients with worse clinical outcomes in SNFgroup2. Hypermethylation of p73, a direct target gene of YAP, has been shown to be strongly correlated with sensitivity to alkylating agents and poor prognosis in patients with MDS [62]. As for the Wnt Signaling pathway, the hyperactivity of this pathway is known to contribute to EMT in human cancer [63].

Presently, the exact role of methylation in gene expression is unknown. A recent pan-cancer study suggested conflicting methylation–expression correlations exist beyond the commonly accepted associations between the promoter region methylation and silencing of gene expression [64]. We observed different patterns of methylation–expression correlation (positive or negative) in our study, suggesting a context-dependent association between gene expression and methylation (Figure 6B). EGFR and SATB2 for example, the two MRs that we identified and show significant correlation with methylation level, have different correlation trends.

## 4. Discussion

Prior studies in osteosarcoma, either using single- or multi-platform data, have focused on identifying driver or marker genes. Recent research shows that, even though high levels of structural variations and mutations cluster in osteosarcoma and contribute to the genomic heterogeneity, few of these recurrent alterations detected from single-platform data can be related to clinical outcome [65]. On the other hand, regardless of the increasing study of marker gene identification from multi-omics data, there is limited research translating the genomic profiles into clinical diagnostics in osteosarcoma. Genotyping and assessing tumors based on not only multi-platform data, but clinical outcomes, is therefore critical to explore novel marker genes and to provide additional guidelines for patient stratification in osteosarcoma.

Combining SCNA, gene expression and methylation through SNF, our integrated and multi-platform investigation reveals three molecularly distinct and clinically relevant subtypes in OS tumors which appear to exhibit differing degrees of aggressiveness or responsiveness to current standard-of-care treatments. The subgrouping criteria was validated using another dataset. Then, we investigated the differences among these three subgroups based on single-platform profiles, and identified several novel factors that potentially contribute to the variation in clinical outcomes at the genomic, transcriptomic and epigenetic level. There is moderate overlap at the individual gene level among these single-platform profiles (only 21 genes intersected among them). Among the top ten MRs identified by MARINa, only two genes (EGFR and NCAM1) have abnormal copy number for at least 30% of tumor samples. Moreover, our single-platform analysis based on the subgrouping reveals different potential marker genes that are associated with clinical outcomes. Among all the potential gene signatures, BRD4, CDK6 and EGFR have been tested in pre-clinical models and their corresponding inhibitors can serve as strong candidates for clinical testing in the future. Lastly, as discussed before, DNA expression and copy number are largely positively correlated, but with a varied degree of correlation from gene to gene. The methylation contributions to transcriptional regulation occur in a complex and dynamic manner. Cross-platform analysis is therefore crucial, especially for investigating complex diseases such as osteosarcoma, to capture multi-layered connections and complexity from different sources.

The osteosarcoma risk assessments provided by this study have the potential to become clinically relevant classifiers. For example, patients with SNFgroup2 profiles experienced the worst clinical outcomes and could be stratified at diagnosis as high-risk patients that warrant more aggressive chemotherapy treatment or prioritization for trials evaluating novel therapies. Indeed, our analysis suggests several targetable pathways that may be important to the biology of some SNFgroup2 tumors. This “wide net” approach may overcome some of the challenges presented by the well-known intra-tumoral heterogeneity of osteosarcoma, which has historically made it challenging to find prognostic biomarker genes that are suitable for universal application. Determining the true utility of such strategies will require additional studies, including prospective analysis in a clinical trial setting.

Going beyond this obvious clinical application, these integrated multi-omics findings in the clinical setting present opportunities to guide targeted drug discovery and validation using cell lines or animal models. Genome-wide and focused screening studies (in vitro or in vivo) facilitated by CRISPR and other techniques may be especially powerful next steps with potential to accelerate the process of novel target identification. It would be reasonable to expect that some of the marker genes identified in this analysis will prove to be genuine drivers of aggressive disease in osteosarcoma.

## 5. Conclusions

Our integrated, multidimensional investigation into osteosarcoma divided osteosarcoma tumors into three subgroups with distinctive molecular features and clinical outcomes. These data provide a comprehensive genomic architecture for osteosarcoma and emphasize the need for data integration from different platforms. Going forward, developing clinically relevant classifiers, and discovering targeted drugs using cell lines or animal models will lead to more effective cancer treatments for osteosarcoma.

## Figures and Tables

**Figure 1 cancers-15-04970-f001:**
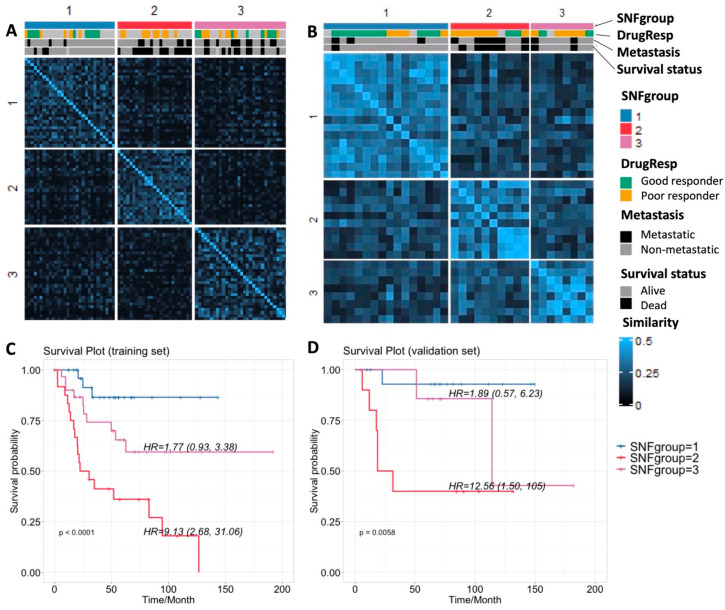
(**A**,**B**) Sample similarity and subgrouping after SNF with clinical annotation for the training and validation sets, respectively; (**C**,**D**) Survival plots for the three subgroups for training and validation sets, respectively.

**Figure 2 cancers-15-04970-f002:**
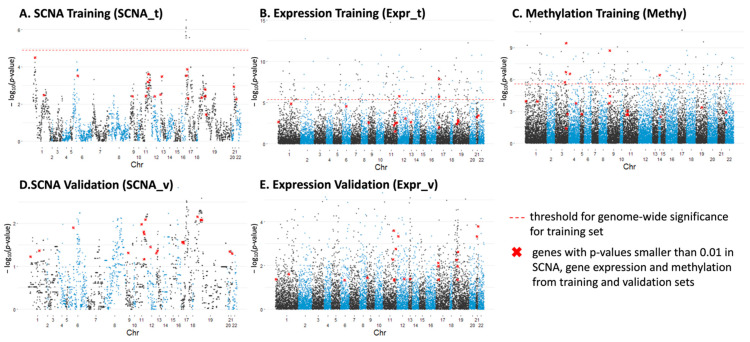
Manhattan plots displaying significantly varied individual genes among three osteosarcoma subtypes along chr1–22 for: (**A**) SCNA from the training set; (**B**) gene expression from the training set; (**C**) methylation from the training set; (**D**) SCNA from the validation set; (**E**) gene expression from the validation set.

**Figure 3 cancers-15-04970-f003:**
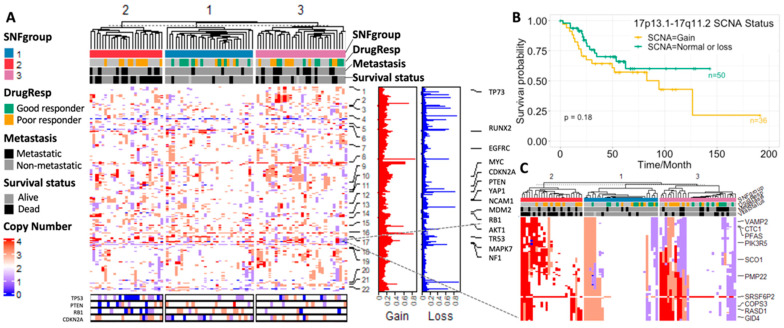
(**A**) SCNA profiles based on the SNF subgrouping from the TARGET data set; (**B**) survival plot for patients with or without copy number gain at 17p13.1–17q11.2; (**C**) SCNA profiles of the 17p13.1–17q11.2 region from the TARGET data set.

**Figure 4 cancers-15-04970-f004:**
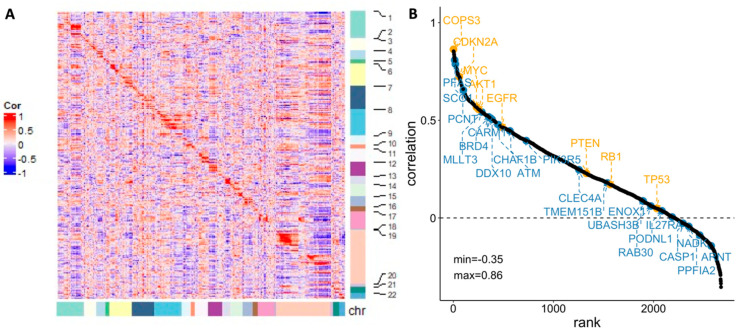
(**A**) Pearson’s correlation between gene expression and SCNA for individual genes (sorted by chromosome location); (**B**) Rank views of correlation between gene expression and SCNA with well-known oncogenes (highlighted in orange, above the line) and potential marker identified by this study (highlighted in blue, below the line).

**Figure 5 cancers-15-04970-f005:**
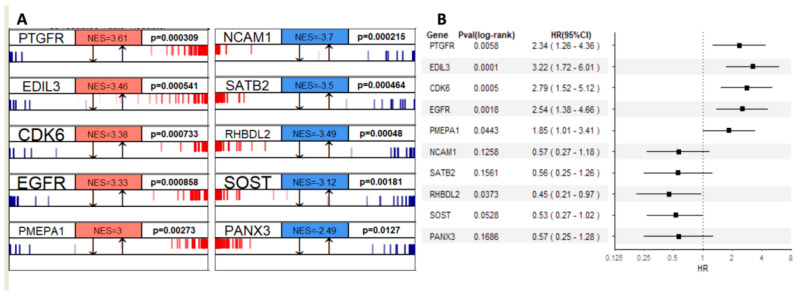
(**A**) Top five master regulators that are activated (red) or repressed (blue) in SNFgroup1 as compared with SNFgroup2 and the corresponding *p*-values; (**B**) The effects of master regulators’ expression on mortality showed by the forest plot.

**Figure 6 cancers-15-04970-f006:**
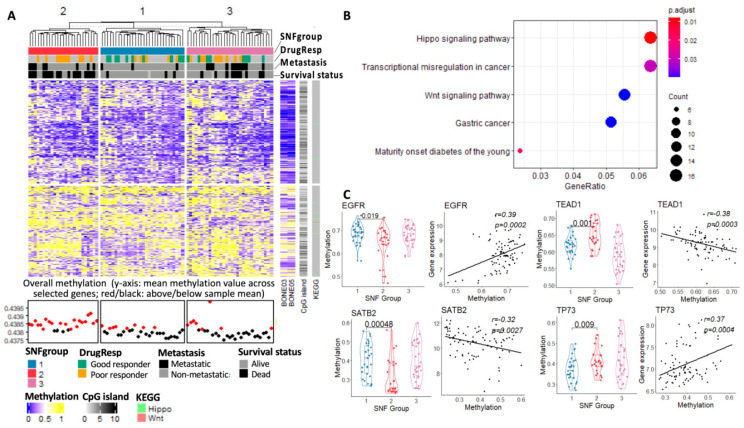
(**A**) Methylation profiles of the top 1% of variable genes in the SNF subgroups from the TARGET data set; (**B**) KEGG enrichment analysis of the genes with *p* < 0.05 from the ANOVA test on the three SNF subtypes; (**C**) methylation value distinction of specific genes among the three SNF subgroups and the corresponding correlation between methylation and expression.

**Table 1 cancers-15-04970-t001:** Patient demography.

Characteristic	n (%)	TARGET (Training, n = 86)	Dr. Korsching (Validation, n = 38)
Median age at diagnosis in year (range)		15 (3–33)	16 (8–60)
Gender	Male	50 (58%)	25 (66%)
	Female	36 (42%)	13 (34%)
Race	White	51 (60%)	Not provided
	Black	8 (9%)	
	Asian	7 (8%)	
	Unknown/not provided	20 (23%)	
Survival status	Alive	54 (63%)	27 (71%)
	Dead	30 (35%)	11 (29%)
	Not provided	2 (2%)	-
Pathologic response to neoadjuvant chemotherapy	Good	18 (21%)	14 (37%)
	Poor	24 (28%)	18 (47%)
	Not provided	44 (51%)	6 (16%)
Metastasis	Metastatic	22 (26%)	15 (39%)
	Non-metastatic	64 (74%)	23 (61%)

**Table 2 cancers-15-04970-t002:** Subgrouping performance evaluated by *p*-values from different methods and data sources.

Method and Data Sources		SNF SCNA + Expr + Methy	Clustering SCNA Only	Clustering Expr Only	Clustering Methy Only
***p*-values**	Survival (log-rank)	2.8905 × 10^−5^ *	0.011 *	0.0035 *	0.0055 *
	Drug response (Fisher’s exact)	0.0039 *	0.1421	0.4635	0.0388 *
	Metastasis (Fisher’s exact)	0.050 *	0.0712	0.4397	0.2712

* *p*-value < 0.05 as statistically significance.

## Data Availability

The datasets analyzed in this study are available in the following repositories. Data from the Therapeutically Applicable Research to Generate Effective Treatments (TARGET, https://ocg.cancer.gov/programs/target, accessed on 31 October 2020) initiative are available from dbGAP via accession phs000218.v22.p8.c1. SCNA and gene expression data for the validation set are available from the ArrayExpress Archive of Functional Genomics Data via accession E-MTAB-3034 (https://www.ebi.ac.uk/arrayexpress/experiments/E-MTAB-3034, accessed on 10 January 2021) and A-AFFY-142 (https://www.ebi.ac.uk/arrayexpress/arrays/A-AFFY-142/?ref=E-MTAB-1215, accessed on 10 January 2021), respectively. Bone samples’ methylation data is available from the Gene Expression Omnibus (GEO) via accession GSE49711 (https://www.ncbi.nlm.nih.gov/geo/query/acc.cgi?acc=GSE49711, accessed on 1 December 2020). The drug sensitivity data is available from Genomics of Drug Sensitivity in Cancer (GDSC, version 17.3, https://www.cancerrxgene.org/downloads/drug_data, accessed on 30 November 2021). Source codes for data process and visualization are available from GitHub at: https://github.com/Shantang3/Osteosarcoma-Multi-omics-Analysis (accessed on 1 September 2023). The processed data during the current study is available from the corresponding author on reasonable request.

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
