# Peer review of "Osteosarcoma Multi-Omics Landscape and Subtypes"

_cancers, 2023, doi:10.3390/cancers15204970_

Round 1

Reviewer 1 Report (New Reviewer)

The manuscript “Osteosarcoma Multi-omics Landscape and Subtypes” is well written by Tang et al. I have a few comments-

1- The author didn’t cover the latest literature, i.e., Southekal et al. Molecular Subtyping and Survival Analysis of Osteosarcoma Reveals Prognostic Biomarkers and Key Canonical Pathways. Cancers 2023, 15(7), 2134. This paper also used Osteosarcoma Multi-omics data from TARGET and performed molecular subtyping. The author must cover this research article and discuss how the current work is different from Siddesh et al. work.

2-      What is the significance of SCNA and gene expression correlation analysis? Do they are not expecting an association between genes’ copy number and expression, unless there is silencing of extra copies of genes?

3- The author didn't discuss Fig. 4b in the manuscript. They must discuss it in the result/discussion section or exclude it from the main manuscript.

4-      There is no uniformity in the references, some of them are in bold while others are not. The author must use the same format across the manuscript.

5-      GitHub link of the source code is broken. The author must fix it in the revised manuscript. Provide more details about these codes in GitHub Readme.

6-      At line number 40 (page 1) there is no space between text and parenthesis “(SCNA).

7-      In figures don’t use underscore for subgroup names. It’s a personal choice but it doesn’t look good.

8- The methodology part is not clear, especially Gene Expression, and DNA methylation part.

The author must do editing of the manuscript in the revised manuscript. Method par is not well written, reframe it.

Author Response

Reviewer 2 Report (New Reviewer)

The authors present an interesting and well-structured manuscript. However, it is necessary to address several issues:

1. The authors must justify the novelty of the biological model presented in the introduction of the state of the art.

2. The methodology section must include a statistical justification of the sample size. The statistical potential must be calculated.

3. The authors must justify the statistical tests used at all times.

4. The authors should improve the quality of the figures extensively and should be more self-explanatory.

5. The figure legends need to be improved.

6. Figure 1 is a bit difficult to interpret and confusing. Please, put a figure of better quality and more representative.

7. The translation of this model should be better discussed. Authors must include the following references:

-doi:10.3390/ijms232314939

Minors points 

Author Response

Reviewer 3 Report (New Reviewer)

Thank you for the authors for the corrected version.

For the previous reviewers comments answers:

1) Last sentence of the abstract seem to be simplified to be easily understandable.

2)Typo is corrected.

3) Table 1 is otherwise corrected, but for the vital (or survival status) should be some extra information at least that the follow-up time is not available. Assessing from the Figures 1C-D and 3B you should have some information about time points, because you could calculate the survival curves. Thus, this answer is not acceptable.

4) See above.

5) Text is now readable.

6) Could there also be notification for different y-scales? Otherwises corrections acceptable.

7) Figure is now readable.

8) I guess this will refer to figure 8. In this figure and the text is now 30%, thus corrected.

9) This answer is not understood correctly. Line 211 and figure 4 text is now said potential maker, instead of potential marker gene(s) as the reviewer required.  In addition line 310 is said potential marker, please check if this should be potential marker gene. And in line 213 "potential marks" is also suspect of wrong text.

10) Figure text is still unclear, as both the red and blue dots are on the line, not under or above. Please clarify. Y-axis should have more numbers: Is the lowest line -0.5?

11) Figure 5 is now readable.

12) Y-axis has now label.

Reviewer 2:

1) Of the 3 mentioned references, only 1 is listed in the paper. There was no texts added in the article. I would prefer to have this explanation also in the paper.

Depending on the Editors opinion, is it possible to correct some minor defiencies, here is a list of those:

-Table 1, instead of Vital status, Survival status would be more appropriate term.

-Figure 1. The usual X-axis for survival would be months/years instead of days. Please correct.

-Table 2, in the title is missing that these are p-values. The limit for significant value is missing. The first column is bolded for unknown reason. The statistically significant values should have some marking.

-Line 143: Please add explanation, that plots are in figure 2.

-Line 167 and line 211: It is unclear where 3.1 is referring, please clarify the text.

-Line 169: If I understand the text correctly, usually this (fraction) is written as percentages: 17% to 52%, please correct the text to be more readable.

-Line 182. Please clarify and simplify the sentence beginning from here: Given the...

-Line 190: Please clarify the head cancer: Head and neck cancer or brain tumor?

-Figure 3B: x-axis now unreadable small. Years/months would be preferred instead of days to be understandable. Please add number of samples to this figure.

-In my opinion the text is not clear in order. Results, discussion, methods and the basis for chosen methods are mixed when they should be clearly separated. 

-Figures are not referred in chronological order in the text= For example figure 4 is before Fig 3.

-Line 237: what is meant by A-AFFY-142?

-Figure text 5B: HR is now mentioned to be survival, but in the text above it is said, that for example CDK6 is associated with worse survival. Please check what is actually meant in this figure: mortality? 

-Figure text 5A: Please write master regulator instead of MR to clarify the text.

-LIne 315: more complex compared to what?

Multiple minor defiencies: word and is missing from line 41. Spaces are missing before the number of references in the text, for example line 27: forms[1]. Line 237 chord might be cohort? Line 255 word "that" gefinitib... is missing. LIne 262: that might be than? And relating correlating? Line 282: hyperactive is hyperactivity? Line 289: Fi ->Fig. 

Probably not all the typographs are here listed. Language is sometimes too complex and for readers more simplified sentences are better.

Round 2

Reviewer 3 Report (New Reviewer)

Thank you for the corrections, the paper was clearly improved.

However, some clarifications still remain to be answered.

Table 2: Clustering Methy only p-value 0.0055 is missing *-sign.

Lines 188 and 231: The Result 3.1 is still unclear where it refers. I found only results 1, 2 and 3.

Figure text 4A. The added text seems to be meant to be figure text 4B, not 4A. In text 4B is still mentioned RED oncogenes, but there are no red text or points seen. 

Line 73: The sentence A Network module... is not clear, please clarify it.

Line 75: outcomes has shown... not correct English.

Lines 77 and 79: The long sentence is referring twice to Siddesh and Pekarek's studies or study? Please clarify and add the reference number.

Line 80: is application meaning implementation?

Line 204: should it be illustrating?

Line 256: is the word "from" unnecessary?

Typographs still found: Please check all "value" words, as p-value is written in multiple ways. Line 261: cohord->cohort. Line 280: Metastatic->metastasis.

Author Response

Table 2: Clustering Methy only p-value 0.0055 is missing *-sign.
A: We added the *-sign.

Lines 188 and 231: The Result 3.1 is still unclear where it refers. I found only results 1, 2 and 3.
A: In the Result section, we have Result 3, and under Result 3 there is 3.1. We changed Result 3.1 to Result 3 – 3.1, hope it helps to clarify.

Figure text 4A. The added text seems to be meant to be figure text 4B, not 4A. In text 4B is still mentioned RED oncogenes, but there are no red text or points seen.
A: Thanks for pointing it out. We’ve unified all the colors reference and changed the figure text.

Comments on the Quality of English Language

Line 73: The sentence A Network module... is not clear, please clarify it.
A: We modified the sentence as “A network assembling well-known and lesser-known candidates for osteosarcoma was constructed by Poos et al. from only SCNA and expression (no methylation).”

Line 75: outcomes has shown... not correct English.
A: We’ve corrected it into “outcomes have shown…”

Lines 77 and 79: The long sentence is referring twice to Siddesh and Pekarek's studies or study? Please clarify and add the reference number.
A: We’ve clarified as “In Siddesh and Pekarek’s studies as mentioned above, there was no validation of the network module or subtyping criteria, and no discussion about clinical outcome association in both studies.” Reference number was also added.

Line 80: is application meaning implementation?
Line 204: should it be illustrating?
Line 256: is the word "from" unnecessary?
A: We’ve corrected all these wording issues.

Typographs still found: Please check all "value" words, as p-value is written in multiple ways. Line 261:
cohord->cohort. Line 280: Metastatic ->metastasis.
A: We’ve corrected all typos. 

This manuscript is a resubmission of an earlier submission. The following is a list of the peer review reports and author responses from that submission.

Round 1

Reviewer 1 Report

In the manuscript titled, "Osteosarcoma Multi-omics Landscape and Subtypes" by Tang et al., the authors use a novel bioinformatic tool similarity network fusion (SNF) to combine copy number analysis, gene expression data, and DNA methylation data into a single output that identified three different groups of osteosarcoma. The molecular profiles of the three subgroups were correlated to clinical outcomes such as drug response, survival and metastasis. While three subgroups were identified, really only group two was the one that became a major focus for clinical relevance. Also, the clinical outcomes of drug response, survival and metastasis are so interconnected pathophysiologically that if one is significantly correlated it is highly likely that the other parameters will be too. The authors did use univariate Cox hazard ratios, but more applicable would be a multivariate analysis. The multiomics approach to subtyping osteosarcoma is very much needed and the training data set translated quite well to the validation cohort, which for a cancer that is as heterogeneous as osteosarcoma, this is a significant accomplishment. To further improve the presentation of the data, I recommend the following revisions: 

1) Simplify the final sentence of the abstract. 

2) Typo in the introduction - change 1908s to 1980s

3) Make changes to Table 1 - 24% for females should be 34%; be consistent by including percentages for all values; for vital status what is the follow up to determine alive vs dead?; for the response to chemotherapy, why are so many values missing? Are they all unknowns? 

4) The vital status really is dependent on the consistent follow up applied to all patients. Typical follow up data is 5-year survival. 

5) Figure 1B - The survival curves are too small. Double the size so the font is legible and move under A and B. 

6) Figure 2 - Keep the y scale consistent between the training and validation cohorts. Also, it was unclear why the dotted threshold was placed on some and not others and even more confusing was the presence of the 21 genes (red dots) below the threshold line. 

7) Figure 3 - Some of the text was too small to read at 100%. 

8) Inconsistency between the text and the figure legend about whether the threshold for expression or SCNA was 25% or 30%. 

9) Page 7 and 8, the authors talk about potential maker genes, did they mean "marker"?

10) Figure 4B requires a better explanation about how the vales of correlation and rank are computed. Does a correlation below 0.0 indicate that an inverse relationship exists? Something to address in the results or discussion section. 

11) Figure 5 is too small. It is very difficult to interpret at 100%. The data was analyzed at 300% and it looks good, but it shouldn't require that level of magnification. 

12) Figure 6A - the overall methylation analysis with the red dots and black dots does not have a label for the y axis. 

Reviewer 2 Report

The authors evaluate gene expression, methylation and cnv from osteosarcoma samples and then combine the data to generate subgroups of patients that have differential survival. The authors state that combining the data provides better predictive capability than any one source.  The authors then describe transcriptional modules generated from RNA-Seq data that is also available from the same sample set.   It is not clear from the paper what the quality of these datasets are in comparison to the TCGA gold standard datasets.  It is also not clear why the authors are only looking at only the most highly changed values within these datasets and suggests that the results maybe highly biased as a result. It is also not clear how the individual datasets compare to previously published analyses of Osteosarcoma copy number change, transcription or methylation (of which their are many). Without knowing the quality /and consistency of the individual datasets relative to previous studies the validity of these finding cannot be meaningfully assessed.  Without further description of the biology of these patterns individually the quality of this paper cannot be assessed.